# Preparation of Highly Substituted Sulfated Alfalfa Polysaccharides and Evaluation of Their Biological Activity

**DOI:** 10.3390/foods11050737

**Published:** 2022-03-02

**Authors:** Zhiwei Li, Yuanhao Wei, Yawen Wang, Ran Zhang, Chuanjie Zhang, Caixing Wang, Xuebing Yan

**Affiliations:** 1College of Animal Science and Technology, Yangzhou University, Yangzhou 225009, China; lzw19951203@outlook.com (Z.L.); weiyuanhao1911@163.com (Y.W.); wangyw42@163.com (Y.W.); zrbswsn@126.com (R.Z.); chuanjiezhang@yzu.edu.cn (C.Z.); 2College of Chemistry and Chemical Engineering, Yangzhou University, Yangzhou 225000, China; 3Joint International Research Laboratory of Agriculture and Agri-Product Safety, The Ministry of Education of China, Yangzhou University, Yangzhou 225009, China

**Keywords:** sulfated modification, response surface methodology, alfalfa polysaccharide, molecular structure, bioactivity

## Abstract

Alfalfa polysaccharides (AP) receive wide attention in the field of medicine, because of their anti-inflammatory property. However, AP has high molecular weight and poor water solubility, resulting in low biological activity. We wanted to obtain highly bioactive alfalfa polysaccharides for further research. Herein, we successfully synthesized highly substituted sulfated alfalfa polysaccharides (SAP) via the chlorosulfonic acid (CSA)-pyridine (Pyr) method, which was optimized using response surface methodology (RSM). Under the best reaction conditions, that is, the reaction temperature, time, and ratio of CSA to Pyr being 55 °C, 2.25 h, and 1.5:1, respectively, the maximum degree of substitution of SAP can reach up to 0.724. Fourier transform infrared spectroscopy also confirmed the existence of sulfonic acid groups on SAP. Despite the increased average molecular weight of SAP, its water solubility is improved, which is beneficial for its biological activity. Further in vitro results showed that SAP exhibited better antioxidant activity and antibacterial ability than AP. Besides, the former can efficiently enhance the viability of oxidatively stressed intestinal epithelial cells compared with the latter. Furthermore, SAP has the potential to inhibit obesity. It is concluded that sulfation modification could improve the antioxidant, antibacterial, bovine intestinal epithelial cells’ proliferation-promoting, and the obesity inhibition abilities of AP. The improvement of AP biological activity may provide references for the utilization of plant extracts that have weaker biological activity.

## 1. Introduction

Plant extracts are important components of functional foods. Plant polysaccharides are common plant extracts, usually obtained by hot water extraction, alkali or acid extraction, enzyme extraction, and microwave extraction [1,2]. Plant polysaccharides are of great interest to many researchers due to their various biological activities [3]. For example, polysaccharides isolated from astragalus polysaccharides have shown significant immunomodulatory effects and anticancer activity [4], and lycium polysaccharides possess anti-aging, anticancer, and neuroprotection properties [5,6,7]. To date, plant polysaccharides have been a research hotspot.

The water solubility and biological activities of polysaccharides are closely related to the molecular size, monosaccharide composition proportion, glycosidic bond characteristics, and other chemical structural characteristics [8,9]. Usually, these properties can be enhanced by structural modification. For example, the water solubility of polysaccharides could be increased by acetylation modification and carboxymethylation modification, and thus its augmented biological activity indirectly [10,11]. Besides, the antioxidant activity of polysaccharides and the body’s immune capacity can be leveled up by phosphorylation [12,13]. It is worth noting that the sulfation modification could not only enhance the water solubility, but also enhance the antioxidant properties of polysaccharides [14]. Furthermore, the physicochemical properties of polysaccharides are related to the content of the substituent group [15,16,17], because polysaccharides are low water-soluble macromolecules. Therefore, it is essential to enhance its water-soluble and biological activity by improving the content of substituents. 

Alfalfa polysaccharides (AP) are active substances that are extracted from alfalfa (*Medicago sativa* L.), which is an important herbaceous plant of the legume family, and widely cultivated in various parts. AP is a macromolecular substance with a variety of biological activities. Several studies on AP molecular structure and biological activity have been reported, which possess antioxidant, immunomodulatory activities [18,19,20,21]. However, the activity of AP is relatively low, which leads to an obstacle in the utilization of AP. Sulfation modification has long been considered a simple and effective method to enhance biological activity, whereas sulfated modification of AP studies have not been reported. In order to improve the utilization of AP, we used chemical methods to increase the activity of AP. The changes in molecular structure and biological activity before and after the modification were compared. The technical route is shown in Figure 1. We hope to provide a reference for others to study AP or similar plant extracts. Herein, we firstly report the preparation of highly substituted SAP via the chlorosulfonic acid-pyridine method, and systematically investigate the optimal conditions by response surface methodology (RSM). The optimal reaction condition was selected based on the results of RSM, that is, the reaction temperature, time, and ratio of chlorosulfonic acid to pyridine were 55 °C, 2.25 h, and 1.5:1, respectively. Under these conditions, highly substituted SAP was prepared, with the maximum degree of substitution of 0.724. We further compared the differences of AP in water solubility, average molecular weight, and monosaccharide composition before and after the sulfation. In parallel, we also measured and compared their main biological activities, including antioxidant activity, antibacterial ability, the viability of oxidatively stressed intestinal epithelial cells, and antiobesity activity. The results proved that the SAP showed a great improvement compared to the AP. The SAP exhibits good antiobesity and antioxidant activities that may allow it to play a role in the treatment of chronic inflammation due to obesity.

## 2. Materials and Methods

### 2.1. Materials and Reagents

The fresh mature alfalfa was harvested in the experimental field of Yangzhou University (Yangzhou, China). The alfalfa was dried at 105 °C in an oven for 15 min, then dried at 60 °C for 12 h. Finally, all alfalfa were crushed and treated by sieving on 100 meshes. Dialysis bags were purchased from Shanghai Acmec Biochemical Co., Ltd. (Shanghai, China). CSA was purchased from Sinopharm Chemical Reagent Co., Ltd. (Shanghai, China). Pyr and N-N dimethylformamide were purchased from Shanghai Macklin Biochemical Co., Ltd. (Shanghai, China). The standard monosaccharide was purchased from BoRui Saccharide Biotech Co., Ltd. (Yangzhou, China). All other chemicals used were of analytical grade.

### 2.2. Preparation and Purification of AP

The alfalfa was dried and pulverized. With slight modifications, the method of Zhang was used in this study [18]. The crude AP was extracted with distilled water by the following extraction conditions: water: powder ratio, 8:1; extraction time, 4 h; extraction temperature, 100 °C. Then, the extraction solution was centrifuged to separate the supernatant, precipitated by adding a 3-fold supernatant volume of 95% ethanol overnight at 4 °C. After 12 h, precipitated AP was collected and redissolved in distilled water. The reconstitution solution was deproteinized by adding a 0.5-fold reconstitution solution volume of 3% trichloroacetic acid (TCA). The supernatant was collected after 12 h. The supernatant was filled in a dialysis sack (molecular weight cut-off 8.0 kDa), and dialyzed in ultrapure water for 48 h. The ultrapure water was replaced every 6 h. It was then precipitated again with 95% ethanol overnight. The supernatant was discarded, and the precipitate was collected. Finally, the precipitate was frozen at −80 °C, then lyophilized in a vacuum freeze dryer, and the dried AP was obtained.

### 2.3. Sulfation of AP and Optimal Test Design of Sulfation

The SAP was prepared by the CSA-Pyr method. Firstly, 10.0 mL of esterification reagent was prepared in a set ratio of CSA and Pyr. In this case, Pyr was added to the three-necked flask, and then the CSA was added drop by drop into the three-necked flask with agitation in the ice-water bath. Then, 200 mg of AP was suspended in 20.0 mL of N-N dimethylformamide at room temperature, and the suspended solution was carefully added to the esterification reagent. After reaching the reaction time, 100.0 mL of ice water was added to the solution. The pH of the reaction solution was adjusted to 7–7.5 with 4.0 mol/L NaOH. The mixture was precipitated with 4 times the volume of 95% ethanol. Finally, the deposits were redissolved in water, and dialyzed (molecular weight cut off 8.0–14.0 kDa) by flowing water for 48 h, then dialyzed by ultrapure water for 24 h. After dialysis, SAP was obtained through lyophilizing, and the SAP was kept in sterile, dry tubes.

The effects of different reaction times (0.5–3.5 h), temperatures (30–90 °C), and CSA/Pyr ratios (8/1 to 1/8 (mL/mL)) on the degree of substitution (DS) of the sulfate group were evaluated. The optimum conditions for each reaction interval were determined to maximize sulfate group content in the polysaccharides. BBD was employed to statistically optimize the test conditions, and evaluate the main effects, interaction effects, and quadratic effects on the sulfate group content of SAP. Based on the BBD principle, 17 experiments were designed using Design-Expert software to test the ratio of CSA to Pyr, reaction temperature, and reaction time on the sulfate group content of SAP.

As shown in Table 1, the three main factors were chosen for this study. They were designated as X_1_, X_2_, and X_3_. The three factors level were coded +1, 0, and −1 for high, medium, and low values, respectively. The following Equation (1) was used to code variables:y_i_ = (X_i_ − X_0_)/∆X(1)
where y_i_ was the coded value of an independent variable; X_i_ was the actual value of an independent variable; X_0_ was the actual value of an independent variable at the center point; ∆X was the step change value of an independent variable.

Based on the BBD test results, the relationships between the independent variables were correlated to obtain a second-order polynomial model. The model was used to predict the optimal point. The model (2) was as follows:Y = A_0_ + ∑A_i_ X_i_ +∑A_j_ X_i_^2^ + ∑A_ij_ X_i_ X_j_(2)
A_0_ was regression coefficients of variables for the intercept, A_i_ was regression coefficients of variables for the linear, A_j_ was regression coefficients of variables for the quadratic, and A_ij_ was regression coefficients of variables for the interaction terms. X_i_ and X_j_ are independent variables (i ≠ j). The fit of the second-order polynomial model was expressed by the coefficient of determination R^2^. The dimensional and contour maps were generated from the regression models.

### 2.4. Structural Characterization of AP and SAP

#### 2.4.1. Polysaccharide Chemical Composition Determination

Total carbohydrates were determined by the phenol-sulfuric acid colorimetric method [22]. Protein content was measured by the Coomassie brilliant blue G-250 method [23]. The AP and SAP were evaluated for their solubility in water according to the 2010 Pharmacopoeia of the People’s Republic of China.

#### 2.4.2. Degree of Substitution (DS)

The sulfate group content (S%) of the SAP was estimated using the barium chloride-gelatin method [24]. The main steps were as follows: the 108.75 mg of K_2_SO_4_ was dissolved in 100 mL of 1.0 mol/L hydrochloric acid. Volumes of 0, 0.04, 0.08, 0.12, 0.16, and 0.20 mL of K_2_SO_4_ solution were aspirated and added to 1.0 mol/L hydrochloric acid to 0.20 mL, respectively. Trichloroacetic acid (3.80 mL) and 1.0 mL of barium chlorate-gelatin (1.0% barium chlorate, 0.5% gelatin) were added and mixed well. After 15 min, the above solution absorbance values, A1, were measured at 360 nm. Using gelatin solution (0.5%) instead of barium chlorate-gelatin solution, the corresponding solution absorbance values, A2, were measured at 360 nm. The standard curve was drawn with sulfate concentration as abscissa, and the value of A1 minus A2 as vertical coordinates. The 3.0 mg of SAP were hydrolyzed with 1.0 mL hydrochloric acid (1.0 mol/L) at 100 °C for 6 h. After cooling, they were replenished with hydrochloric acid to 1.0 mL. The 0.2 mL sample solution with 1.0 mL barium chloride-gelatin and 3.8 mL trichloroacetic acid was fully shocked and incubated at room temperature for 15 min. The A2 value was measured at 360 nm. The gelatin solution, instead of the barium chloride-gelatin solution, was added into tubes and determined at 360 nm, and then A4 was obtained. The sulfate group content of the sample was calculated from the standard curve. DS was calculated [15] by the following Equation (3):DS = (1.62 × S%)/(32 − 1.02 × S%)(3)

#### 2.4.3. Component Analysis

The monosaccharide compositions were determined by high-performance ion chromatography (HPIC) as described by Sun [25], with slight modifications. Briefly, 10 mg samples were taken into a 5.0 mL ampule, sealed with 10.0 mL of 3.0 mol/L trifluoroacetic acid, and then acidified at 110 °C for 3 h. The solution was further evaporated by nitrogen-blowing, and remixed with 10.0 mL of water. The 100 μL resulting solution was remixed with 900 μL of deionized water, and then centrifuged at 12,000 rpm for 5 min. The supernatant was taken into IC for analysis.

#### 2.4.4. FT-IR Analysis

The dried polysaccharide samples were ground with dried KBr powder, and pressed into thin pellets for FTIR measurements (FT-IR 650 Fourier transform infrared spectrometer).

#### 2.4.5. Molecular Weight Determination

The molecular weight was determined by high-performance gel permeation chromatography (HPGPC), a system equipped with two columns and a differential refractive index detector. The BRT105-104-102 column (8 × 300 mm, BoRui Saccharide Biotech Co. Ltd., Yangzhou, China) was used for the separation of polysaccharides. NaCl solution (0.05 mol/L) was used as the mobile phase at a flow rate of 0.6 mL/min, and the column temperature was maintained at 40 °C. A series of standard dextran solutions with molecular weights ranging from 5.0−667.8 kDa were used to generate the calibration curve.

### 2.5. Assay for Biological Activity

#### 2.5.1. Scavenging DPPH Radical Assay

The scavenging capacity of hydroxyl radical was measured by the method of Yuan, with slight modifications [26]. Different concentrations of polysaccharide solutions were configured (0.3, 0.6, 0.9, 1.2, 1.5, 1.8 mg/mL). DPPH (0.1 mmol/L) was dissolved in anhydrous ethanol. The 2.0 mL of polysaccharide solutions were mixed with 2.0 mL of DPPH. The mixture was shaken vigorously and left to sit in the dark for 30 min. The absorbance of the mixture at 517 nm was measured. The experiment was repeated three times, each with duplicate samples. The VC was selected as a positive control. The percentage of scavenging effect was calculated with the following Equation (4):Scavenging rate (%) = (A0 − (A1 − A2))/A0 × 100%(4)
where A0 is the absorbance of the mixture of DPPH and anhydrous ethanol, A1 is the absorbance of the mixture of DPPH and polysaccharide solutions, and A2 is the absorbance of the mixture of anhydrous ethanol and polysaccharide solutions.

#### 2.5.2. Scavenging Hydroxyl Radical Assay

The scavenging capacity of hydroxyl radical was measured by the method of Nie [27], with slight modifications. The main steps were as follows. Different concentrations of polysaccharide solutions were configured (0.3, 0.6, 0.9, 1.2, 1.5, 1.8 mg/mL). The reaction systems contained 1.0 mL FeSO_4_ (1.5 mmol/L), 0.7 mL H_2_O_2_ (6.0 mmol/L), 0.3 mL sodium salicylate (20.0 mmol/L), and 1.0 mL sample solution. The reaction systems were incubated at 37 °C for 1 h, then were measured at 510 nm. The experiment was repeated three times, each with duplicate samples. The VC was selected as a positive control. The percentage scavenging effect was calculated with the following Equation (5):Scavenging rate (%) = (A0 − (A1 − A2))/A0 × 100%(5)
where A0 is the absorbance of the mixture after water is used instead of polysaccharide solution, A1 is the absorbance of polysaccharide solution, and A2 is the absorbance of the mixture after water is used instead of H_2_O_2_.

#### 2.5.3. Ferric Reducing Antioxidant Power (FRAP)

Antioxidant power was determined by the FRAP method. A FRAP assay was conducted using a modified version of an earlier study [28]. Different concentrations of polysaccharide solutions were configured (0.3, 0.6, 0.9, 1.2, 1.5, 1.8 mg/mL). The reaction solution was prepared to mix acetate buffer (pH 3.6, 300.0 mmol/L); 2, 4, 6-trispyridyl-s-triazine solution (10.0 mmol/L); and FeCl_3_ solution (20.0 mmol/L) with a volume ratio 10:1:1. Different concentrations of VE solution were prepared. Volumes of 10 µL of VE solution and 190 µL of FRAP solution were mixed and then kept at 37 °C in a water bath for 10 min. The absorbance value was measured at 593 nm. The absorbance was used as the vertical coordinate, and the concentration of VE as the horizontal coordinate to produce a standard curve. Afterwards, 190 μL reaction solution and 10 μL sample solution were mixed. Then mixed solution was incubated in a 37 °C water bath for 10 min. The absorbance was determined at 593 nm. The concentration of VE was indirectly expressed, as with the equivalent antioxidant activity of the sample. The experiment was repeated three times.

#### 2.5.4. Antimicrobial Capacity Assay

The 100 μL LB liquid medium containing 1.0 × 10^5^ cfu Staphylococcus aureus (*S. aureus*) was added to each well of a 96-well plate. Then AP or SAP diluted by LB liquid medium were added, and adjusted the concentrations to 0.25, 0.50, 0.75, 1.00, 1.5, 2.00, 3.00, and 4.00 mg/mL, each set with three replicates. The 96-well plate was put in an incubator at 37 °C for 24 h. A plate counting method was used for enumeration of bacteria. Gentamicin sulfate (GS) was used as the positive control group. The experiment was repeated three times, each with duplicate samples.

#### 2.5.5. Cell Viability Assay

The effect of polysaccharides on the bovine intestinal epithelial cells’ (BIEC) viability was examined by CCK-8 assay. They were cultured in DMEM/f12 with 10 % FBS. It was divided into control groups, polysaccharides groups, and sulfated polysaccharides. Firstly, the concentration of BIEC was adjusted to 1.0 × 10^4^ cells per mL, then 200 μL was added to each well. The cell was cultured until 80% were adherent to growth in the incubator at 37 °C containing 5% CO_2_. Next, the original culture medium was discarded, and cells were washed with PBS. The purified polysaccharide samples were dissolved in a medium, and passed through a 0.22 µm aseptic filtration membrane. Afterwards, 200 μL different concentrations of polysaccharide samples were added to their corresponding wells (125, 250, 500, 750, 1000 μg/mL). The blank group only contained medium. Each group had three repeats, and were cultured for 24 h. After 24 h, the plate was taken out, and the medium was discarded. The 100 μL medium with 10% CCK-8 reagent was added to the corresponding wells, and incubated at 37 °C. After 4 h, the absorbance at 450 nm was measured. 

The protective effect of polysaccharides on BIEC under oxidative stress was assayed by CCK-8. Cells were cultured as described above. The control group, model group, SAP group, and AP group were set up. Oxidative damage of the cells was induced using lipopolysaccharides (LPS). The model group was induced for 24 h by 1.0 mg/mL LPS. The SAP and AP groups were incubated with different concentrations of polysaccharides (62.5, 125, 250, 500, 750 μg/mL) and 1.0 mg/mL LPS for 24 h. The control group was incubated for 24 h without the addition of polysaccharides. Then, cell viability was measured by the CCK-8 method. 

#### 2.5.6. Animal Experiments

The female C57/BL6 mice (weighted 15 ± 0.5 g, 4–5 weeks) were supplied by the Laboratory Animal Center of Yangzhou University (Yangzhou, China). All mice were placed in mouse cages, and adapted for one week under controlled conditions (temperature 23 ± 1 °C, lights on 12 h every day). All experiments were performed following the Regulations of Experimental Animal Administration issued by the State Committee of Science and Technology of the People’s Republic of China. After the adaptation period, all mice were randomly divided into four groups (eight in each group), including two experimental groups, as well as one normal control group, and one high-fat diet (60% calories from fat) control group. During the experimental period, the experimental groups were gavaged with AP or SAP (800 mg/kg) once daily. The polysaccharide concentration refers to the literature of Zhang [29], which is a highly concentrated polysaccharide solution. The experiment lasted 30 days. Rat chow was weighed daily during the last week before experimentation to calculate feed intake. On the last day of the experiment, all mice were fasted with water for 12 h, weighed, and their body weight was recorded. Tail vein blood samples were obtained for plasma glucose measurement by a glucometer (ONE TOUCH Ultra, Johnson & Johnson Medical (Shanghai) Ltd., Shanghai, China). Plasma lipids were measured with commercially available kits.

### 2.6. Statistical Analysis

Design and data analysis of response surface tests was carried out using Design-Expert software (version 8.0.0, Minneapolis, MN, USA). The other data were statistically analyzed using the SPSS software (version 13.0, Chicago, IL, USA) by one-way ANOVA and LSD tests. The *p* < 0.05 was regarded as significant. Graphics were produced using Origin software (version 8.0, Northampton, MA, USA).

## 3. Results

### 3.1. Sulfation of Alfalfa Polysaccharides

As evident from Figure 2a, the DS increased with the ratio of CSA to Pyr increasing from 8:1 to 1:1, and the DS decreased with the ratio increasing from 1:1 to 1:8. Under the current experiment conditions (reaction time 2 h, reaction temperature 60 °C), the optimum ratio was 1:1 in this study, and the real optimum ratio should be between 1:2 and 2:1. As shown in Figure 2b, the DS increased as the reaction temperature increased from 30 to 50 °C, and the DS decreased with increasing temperature. Under current experiment conditions (reaction time 2 h, reaction ratio 1:1), the optimum reaction temperature was 50 °C, and the real optimum reaction temperature should be between 40 and 60 °C. It can be seen from Figure 2c that the DS increased as the reaction time increased from 0.5 to 2.5 h. Shorter reaction times would lead to incomplete reactions, but when the reaction time was more than 2.5 h, the degree of substitution showed a decreasing trend. Under current experiment conditions (reaction temperature 50 °C, reaction ratio 1:1), the optimum reaction time was 2.5 h, and the real optimum reaction temperature should be between 2 and 3 h.

### 3.2. Model Fit and Predictive

The BBD was aimed at optimizing the reaction temperature, reaction time, and the ratio of CSA to Pyr. There was a total of 17 runs for optimizing the three individual parameters. The experimental conditions and DS are shown in Table 2. The 3D response surface plots and contour plots are shown in Figure 3.

Figure 3a shows the 3D response surface plot and the contour plot developed with varying ratios of CSA to Pyr and different temperatures at a fixed reaction time (2.26 h). The DS shows an increasing, and then decreasing, trend at different temperatures as the ratio increases. At different ratios, the degree of substitution keeps an increasing trend with increasing temperature until the temperature reaches the central point, and decreases after reaching the central point.

Figure 3b shows the 3D response surface plot and the contour plot developed with varying reaction times and temperatures at a fixed ratio (CSA to Pyr of 1.5: 1). It is observed that the DS increased gradually with the increase of reaction temperature or reaction time. Also, increasing the reaction temperature or reaction time above the threshold level led to the DS deceasing.

Figure 3c shows the 3D response surface plot and the contour plot developed with varying ratios of CSA to Pyr and different reaction times at a fixed reaction temperature (55.48 °C). A similar result in which the DS increased gradually with the increase of ratios of CSA to Pyr or reaction time is observed in Figure 2c. The DS also decreases when the ratio of CSA to Pyr or reaction time is above the threshold level. 

The final model for the prediction of the DS is shown in Equation (6):DS = −4.99664 + 0.094558X_1_ + 0.362778X_2_ + 2.49333X_3_ − 0.000067X_1_X_2_ − 0.0119X_1_X_3_ − 090667X_2_X_3_ − 0.000609X_1_^2^ − 0.051378X_2_^2^ − 0.3756X_3_^2^(6)
where X_1_ is the reaction temperature, X_2_ is the ratio of CSA to Pyr, and X_3_ is the reaction time.

A variance analysis of this model is shown in Table 3. The value of R^2^ was obtained at 0.9849, which indicated that the model could explain experimentally, and predict data. The value of Adj-R^2^ (0.9655) also confirmed the proposed model. The predicting optimum conditions for sulfation modification of the polysaccharides were as follows: reaction time 2.26 h, reaction temperature of 55.48 °C, the ratio of CSA to Pyr of 1.5: 1, and the model predicted a maximum response of 0.715. Based on the actual situation, the experiment rechecking was performed under these modified optimal conditions: reaction time of 2.25 h, the reaction temperature of 55 °C, and the ratio of CSA to Pyr 1.5:1. A mean value of 0.724 (N = 3) was obtained according to the actual experiments, which was signed in agreement with the predicted value. The model was efficient.

### 3.3. Structural Characterization

#### 3.3.1. Chemical Analysis

As showcased in Table 4 and Figure 4, the color of AP changes from white to light yellow after sulfation modification (Figure 5). The sulfation modification effectively improves the solubility of AP. The protein content of SAP was lower than that of AP, indicating that TCA can effectively remove protein from polysaccharides. The carbohydrate content of SAP decreased after sulfation modification.

#### 3.3.2. FTIR Analysis

The FTIR spectra of AP and SAP are shown in Figure 6. The properties of AP and SAP show typical and strong absorption peaks at 3390 cm^−1^ and 3392 cm^−1^ for the −OH stretching vibrations. The absorption bands at 3600–3200 cm^−1^ are the absorption peak of the stretching vibration of −OH, and the absorption peaks in this region are the typical characteristics of polysaccharides. The absorption bands at approximately 2900 cm^−1^ and 2925 cm^−1^ are due to the C−H stretching vibration. The absorption peak at 929 cm^−1^ is caused by the asymmetric ring stretching vibration of the pyran ring. Compared with AP, two typical absorption bands appear in the FTIR spectra of SAP. The SAP has a strong absorption peak at 1258 cm^−1^, caused by asymmetric O=S stretching vibration absorption. In addition, the absorption peak at 809 cm^−1^ is caused by the absorption of tensile vibration of C-O-S. These results indicate that the sulfation of AP was successful.

#### 3.3.3. Average Molecular Weight

There was a linear relationship between the retention time and the logarithm of the molecular weight. The calibration curve formula was as follows: lgMw = −0.1889x + 12.007 (R^2^ = 0.9943), lgMn = −0.1752x + 11.304 (R^2^ = 0.9931). According to the equation, the Mw of AP and SAP were determined to be 2.2 × 10^4^ and 2.5 × 10^4^ Da, respectively. The Mn of AP and SAP were determined to be 1.2 × 10^4^ and 1.7 × 10^4^ Da, respectively.

#### 3.3.4. Monosaccharide Composition

Through acid hydrolysis, AP and SAP were subjected to monosaccharide composition analysis. The results are shown in Figure 7. AP is mainly composed of fucose, arabinose, galactose, glucose, xylose, and galacturonic acid, with a molar ratio of 5.08%, 1.16%, 0.41%, 67.97%, 1.31%, and 24.06%, respectively. SAP monosaccharide composition is the same as AP, with a molar ratio of 2.56%, 2.39%, 0.79%, 79.38%, 1.66%, and 13.19%, respectively.

### 3.4. Biological Activity

#### 3.4.1. Antioxidant Capacity

The antioxidant capacity of AP and SAP is shown in Figure 8. The findings indicate that the scavenging capacity of AP to DPPH radicals is lower than SAP. The hydroxyl radicals scavenging capacity and reducing capacity of AP are also lower than SAP. The scavenging effects of the SAP on DPPH radicals and hydroxyl radicals are lower than that of VC at the same concentration. The antioxidant capacity of the two samples is in a concentration-dependent fashion. 

#### 3.4.2. Antimicrobial Capacity

The ability of AP and SAP to inhibit bacteria is shown in Figure 8d. The results suggest AP could inhibit the growth of S. aureus, which is weaker than SAP at the same concentration. This indicates that the sulfated modification enhanced the bacterial inhibitory property of AP. The inhibition curves of AP and SAP tend to be flat at concentrations higher than 1.0 mg/mL. The bacterial inhibition rate of SAP was less than 50%.

#### 3.4.3. Cell Viability

Figure 9a provides an overview. The viability of BIEC decreases with increasing SAP concentrations (250~1000 μg/mL). The results show that cell viability is significantly increased (*p* < 0.05), which is affected by SAP at the 125 μg/mL concentration. As the AP concentration increased, the viability of BIEC showed an increasing, then decreasing, tendency. The highest cell viability is observed for the concentration of 500 μg/mL. We also examined the effect of the AP and SAP on the BIEC viability of the oxidative stress statuses. As shown in Figure 9b, the SAP group showed an increasing trend in cell viability at the concentration from 62.5 μg/mL to 500 μg/mL. Cell viability tends to decrease after SAP concentrations above 500 μg/mL. The SAP group was significantly higher than the AP group when concentrations were above 500 μg/mL (*p* < 0.05). The cell viability of the AP group increased with increased concentration. The cell viability of the AP group was lower than the SAP group when concentrations were lower than 500 μg/mL, though not significantly.

#### 3.4.4. Animal Experiments

As is visible from Figure 10, the average food intake of the control group is significantly higher than in the rest of the test groups (*p* < 0.05). The body weight in the HFD group is the highest, followed by the AP and SAP group, and the lowest in the control group. All comparisons between groups are significantly different (*p* < 0.05). The highest blood glucose level is seen in the HFD group. The blood glucose level in the AP group is significantly higher than those in the SAP group (*p* < 0.05). There is no significant difference between the control and SAP groups. The differences of TC and TG results in each group are consistent with bodyweight result differences. There is no significant difference in HDL-C between the AP and SAP groups, but it is significantly higher than that in the other two groups (*p* < 0.05). The HFD group has the lowest HDL-C level, and is significantly lower than in the remaining groups (*p* < 0.05). The LDL-C level is also not significantly different between the AP and SAP groups, but remains significantly higher than the control group (*p* < 0.05). The HFD group has the highest LDL-C level, and is significantly higher than in the remaining groups (*p* < 0.05). 

## 4. Discussion

### 4.1. Sulfation of Alfalfa Polysaccharides

The reagent ratio, reaction time, and reaction temperature are important factors that influence the outcomes of DS. The substitution reaction is more likely to occur under strong acid conditions. The substitution reaction may not be sufficient under weak acid conditions. However, polysaccharides are easily degraded when the acidity is too strong. Thus, the degree of substitution tends to decrease when the ratio is higher than 1:1. Substitution reactions are more likely to occur under high-temperature conditions, but polysaccharides are also easily decomposed [30], and are prone to carbonation, too. As such, the degree of substitution tended to decrease when the ratio was higher than 50 °C. Shorter reaction times would lead to incomplete reactions, but when the reaction time was more than 2.5 h, the degree of substitution showed a decreasing trend. This phenomenon might be due to the degradation of polysaccharides induced by the prolonged exposure of polysaccharides to strongly acidic conditions [16].

### 4.2. Structural Characterization

The sulfation modification effectively improves the solubility of AP, which is consistent with the literature that chemical modification can enhance water solubility [10]. Good water solubility is beneficial for the further clinical application of AP. The carbohydrate content of SAP decreased after sulfation modification, consistent with previous studies [15], which might be due to the degradation of polysaccharides by sulfation. 

Compared with the AP, the sulfated derivatives showed an increase in Mw and Mn. Some studies have shown that sulfation modification decreases the molecular weight of polysaccharides [15]. However, some studies have also found that sulfation modification increases the average molecular weight of polysaccharides [31]. This may be related to the degradation of the carbon chain, and the molecular weight of the substituted group. Our results indicated that the main carbon chain of the AP was not easily degraded under the optimized test conditions, and the structure of the AP was not seriously damaged.

A study showed that the AP was composed of fucose, arabinose, galactose, glucose, xylose, mannose, galactose, galacturonic acid, and glucuronic acid. Glucose, glucuronic acid, and galacturonic acid were the three main monosaccharides [18]. Shang found that AP consisted of five monosaccharides: glucuronic acid, rhamnose, glucose, galactose, and xylose. Rhamnose, glucose, and galactose were the three main monosaccharides [20]. In our study, the main three monosaccharides were glucose, fucose, and galacturonic acid. The monosaccharide species and monosaccharide composition ratios of alfalfa polysaccharides were not the same due to the different extraction methods, alfalfa growth periods, alfalfa sample processing methods, and alfalfa species. Compared to the AP, the percentage of fucose and galacturonic acid in SAP decreased, which may be due to the degradation of the carbon chain in the sulfation process [32].

### 4.3. Biological Activity

Antioxidant activity is an important indicator to evaluate the function of natural extracts. DPPH is a stable nitrogen-centered free radical, which presents a purple color in organic solvent, and has a maximum absorption peak at ultraviolet 517 nm [33]. DPPH has a single electron, and antioxidants neutralize free radical properties by transferring electrons or hydrogen atoms to pair with the single electron of DPPH. Hydroxyl radical is an important free radical species with great electron gaining capacity. Hydroxyl radicals can cross freely through the cellular membrane, and enter the cell and attract most biomolecules in cells, such as protein, enzyme, lipids, and DNA. This will trigger the free radical chain reaction, resulting in direct cellular damage [34]. Fortunately, the antioxidant can donate an electron to hydroxyl radicals to block the free radical chain reaction. The reducing capacity of the compound serves as a significant indicator of evaluating its antioxidant activity. FRAP was widely used to assay the antioxidant ability of substances [35]. Polysaccharides usually have negative charges [36,37], which can scavenge DPPH free radicals, or interrupt the chain reaction of free radicals by providing electrons. Our results suggest that AP also has similar antioxidant abilities. Sulfate groups are negatively charged groups. The sulfate groups are linked to the carbon chain of the polysaccharide by sulfated modification. Therefore, the modified AP carried more negative charges, and can provide more electrons to scavenge DPPH free radicals or hydroxyl radicals. Good water solubility was also beneficial to enhance the antioxidant properties of SAP. The sulfation modification also improved the reducing ability of AP, which is consistent with the results of other sulfated modified polysaccharides [26]. The findings showed that sulfation modification can improve the antioxidant activity of AP.

*S. aureus* is a widely spread gram-positive bacterium. *S. aureus* infections cause various inflammatory reactions, and the widespread use of antibiotics in treatment has made aureus resistant to drugs [38]. Currently, some natural extracts have been found to inhibit the growth of *S. aureus* [39,40]. Natural polysaccharides have a weak ability to inhibit the growth of *S. aureus*, though their ability can be improved after chemical modifications [15]. Studies concluded that polysaccharides inhibit the growth of *S. aureus* by altering the permeability of the cell membrane, which causes an outflow of intracellular material, and damages the bacteria [41]. SAP carries a more negative charge, and provides more electrons to increase the negative charge outside the cell membrane. In the case of increased cell membrane permeability, the cell membrane potential of bacteria is disturbed by a high density of negative charge. It may influence the cell membrane to exchange substances. This could be the reason for the stronger antimicrobial capacity of SAP. Our study found that bacteria become resistant to high concentrations of AP and SAP, with less than 50% bacterial inhibition. The AP showed no strong antibacterial ability. Therefore, we concluded that sulfation modification can improve the antibacterial ability of polysaccharides. The bacterial inhibitory ability of the modified polysaccharide was limited by the ability of the raw material itself. This also indicated that the enhancement of bacterial inhibitory ability is related to the sulfate group. Further studies are needed to investigate why bacteria become resistant to high polysaccharide concentrations.

The intestinal tract is a complex organ with functions closely related to food digestion, nutrient absorption, and body immunity. High-fat diets increase intestinal permeability, and promote the release of lipopolysaccharides (LPS) into the blood. This will cause an inflammatory response [42]. Studies have reported that polysaccharides can improve symptoms caused by high-fat diets by modulating intestinal microorganisms [43]. At present, it is not clear whether polysaccharides can improve intestinal inflammation through other pathways. The intestinal epithelial cells are an integral component of the enteric system. Hence, we examined the effect of the AP and SAP on the cell viability of BIEC. Several studies have shown that polysaccharides can promote cell proliferation, and the tolerance of cells to polysaccharides was inconsistent [15,44]. The study of Li [45] showed that low concentrations of polysaccharides could promote cell proliferation, and high concentrations could inhibit cell proliferation. Several studies have also shown that high concentrations of polysaccharides could inhibit the proliferation of tumor cells, and induce apoptosis [46,47]. They inferred that AP or SAP produce cytotoxic effects on cells with increasing concentrations, and this can lead to apoptosis. The effect of AP and SAP on cells in this study was similar to their study. SAP promotes cell proliferation at a lower concentration than AP, suggesting that the cells are more sensitive to SAP. This study found that concentrations of SAP lower than 500 μg/mL could promote the proliferation of BIEC of oxidative stress statuses, which helped to reduce intestinal permeability, and relieve inflammation of the intestinal tract. Compared to AP, SAP showed a better promotive effect of proliferation on oxidatively stressed BIEC, which can be owed to the better antioxidant capacity of SAP. High concentrations of AP or SAP reduced cell activity, which is due to the toxicity of high concentrations of polysaccharides.

Previous studies suggested that long-term consumption of a high-fat diet (HFD) causes obesity [48]. Obesity leads to the development of multiple metabolic disorders. At present, obesity is associated with a wide array of morbidities, including type 2 diabetes, metabolic syndrome, and cerebrovascular diseases [49]. The symptoms of obesity can be alleviated by dieting, surgery, drugs, and exercise. However, bariatric surgery and weight-loss drugs may lead to complications and undesirable side effects. Dieting in obese subjects is typically followed by weight regain [50]. Due to this, the prevention of obesity deserves closer attention than obesity treatment. With more people avoiding chemical drug side effects, there have been many attempts to identify natural products that counteract obesity. Plant polyphenols are common natural products, and studies have confirmed their effect on relieving obesity [51]. Currently, researchers have found that saponins and polysaccharides also have anti-obesity effects, and are actively being studied in further research [52,53]. Our findings showed that prolonged consumption of high-fat diets led to a significant increase in body weight, blood glucose, and lipids in mice. After gavage with high concentrations of AP, all indexes were decreased, and there was no significant change in average food intake, indicating that AP could alleviate obesity, and did not affect appetite. The body weight, blood glucose, and blood lipids of mice gavaged with high concentrations of SAP were close to those of the control group, indicating that SAP alleviated obesity better than AP. Obesity leads to a consequence of a pro-oxidant/antioxidant imbalance [54]. The better antioxidant capacity of SAP could be useful for the rehabilitation of the balance. This may be one of the reasons for the better effect of SAP than AP. However, obesity is influenced by several factors, such as the community structure of the gut microbiota. Further studies are warranted to explore the effects of AP and SAP in vivo.

## 5. Conclusions

In summary, a water-soluble polysaccharide was isolated from alfalfa using the hot water extraction method, and then sulfated by CSA-Pyr. RSM was used to estimate and optimize the experimental time, experimental temperature, and reagent proportions. We obtained a high correlation quadratic polynomial mathematical model by Design-Expert software. The optimal reaction conditions for sulfation were determined as follows: reaction time of 2.5 h, reaction temperature of 55 °C, and a ratio of CSA to Pyr of 1.5:1. Under this condition, the degree of substitution of the polysaccharides was up to 0.724, which agreed closely with the predicted value. The spectra results showed that the sulfation reaction had occurred successfully. The sulfated modification changed the molecular weight and monosaccharide composition. Our results indicated that sulfate modification could improve the antioxidation properties of AP. Sulfate modification enhanced the ability of the AP to inhibit the growth of *S. aureus*. Different concentrations of AP and SAP could promote or inhibit the proliferation of BIEC, and BIEC was more sensitive to SAP than AP. SAP also has the better potential to inhibit obesity than AP. Furthermore, the effects of SAP in vivo will be conducted in further studies.

## Figures and Tables

**Figure 1 foods-11-00737-f001:**
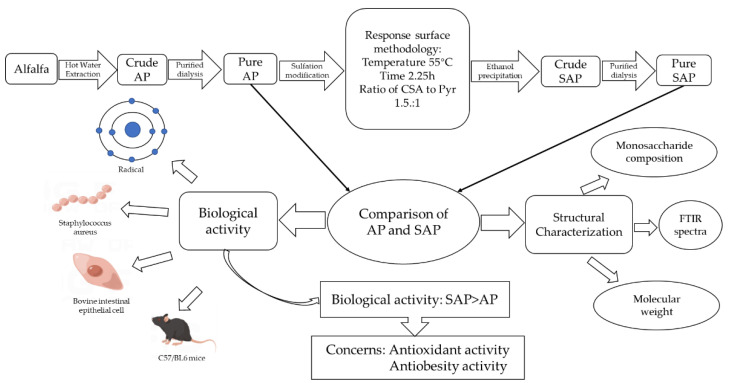
Overview schematic of the study.

**Figure 2 foods-11-00737-f002:**
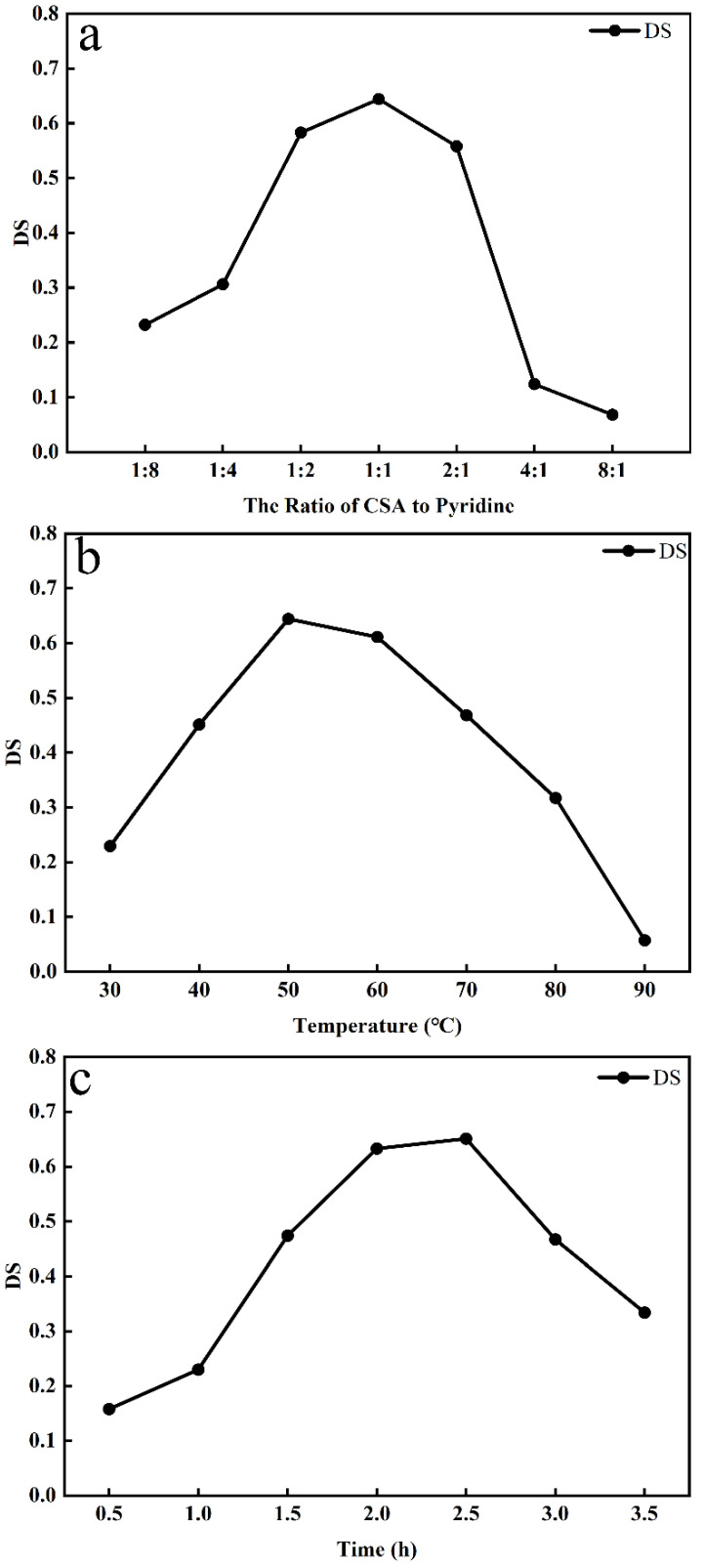
The effects of various parameters on DS: (**a**) ratio of CSA to Pyr; (**b**) temperature; (**c**) reaction time.

**Figure 3 foods-11-00737-f003:**
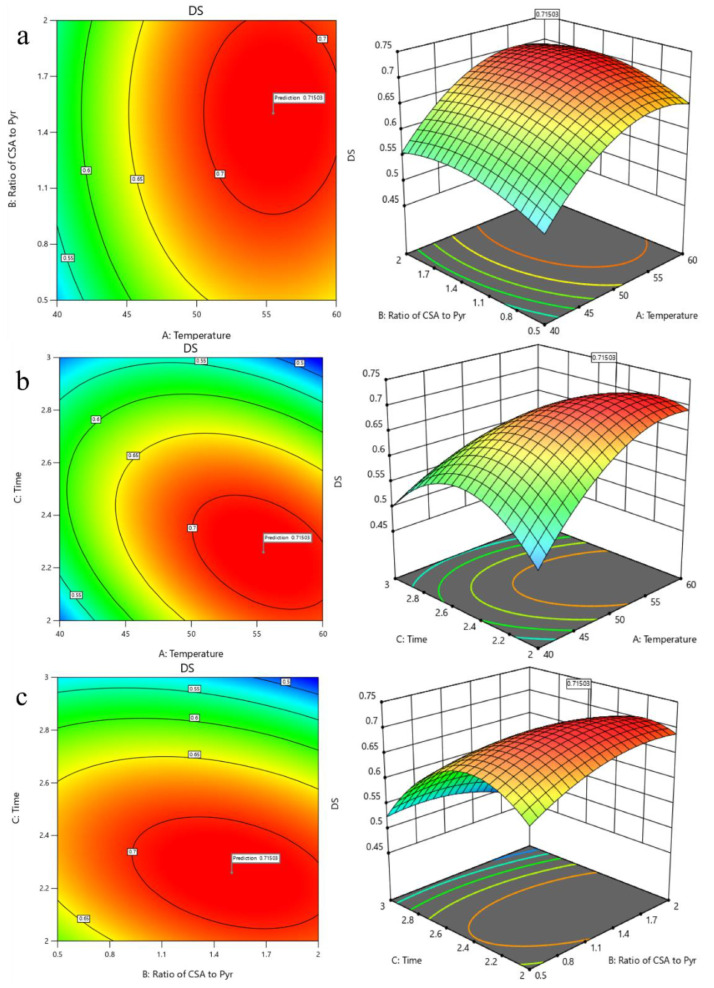
Tri-dimensional response surface and contour plots. (**a**) Temperature and ratio of CSA to Pyr; (**b**) temperature and time; (**c**) ratio of CSA to Pyr and time.

**Figure 4 foods-11-00737-f004:**
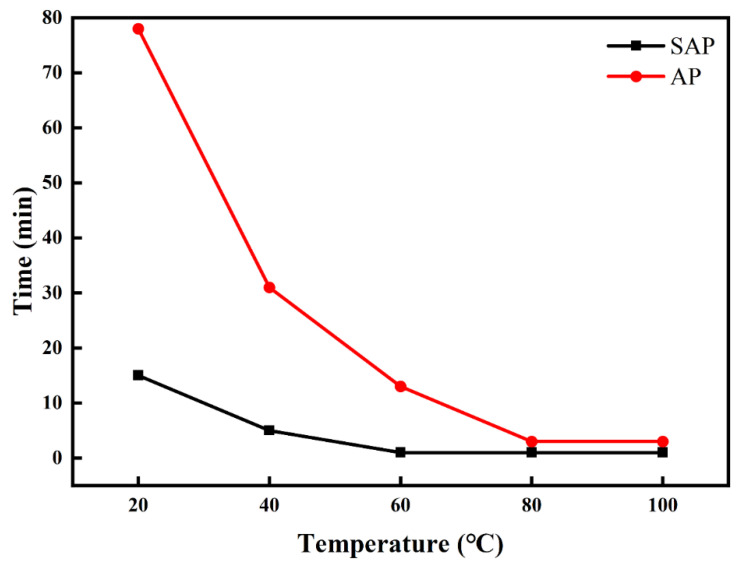
Solubility of AP and SAP.

**Figure 5 foods-11-00737-f005:**
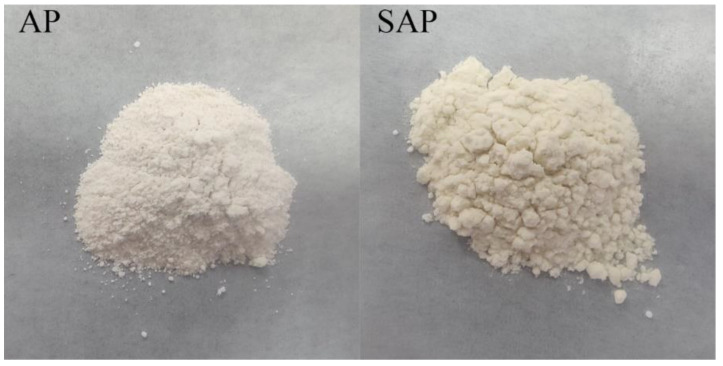
Photograph of AP and SAP.

**Figure 6 foods-11-00737-f006:**
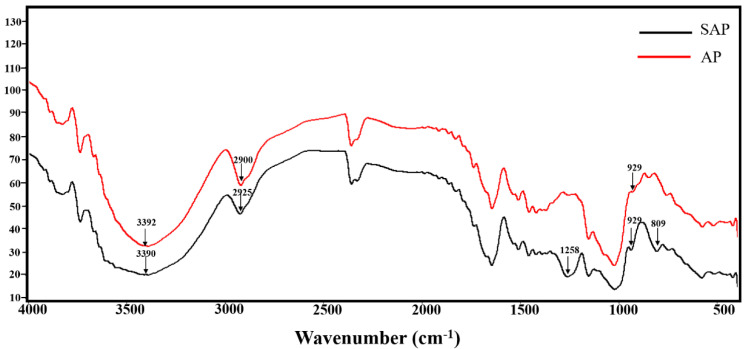
FTIR spectra of AP and SAP.

**Figure 7 foods-11-00737-f007:**
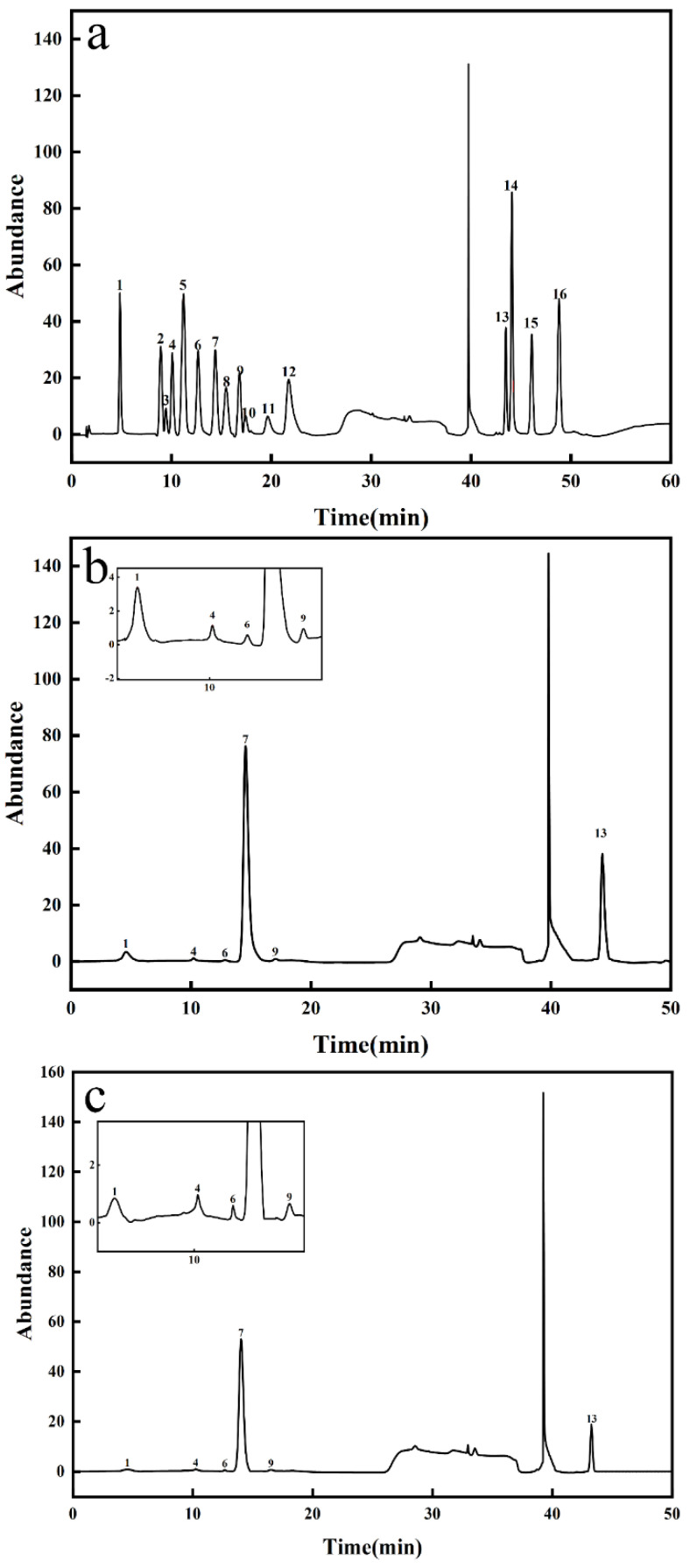
Monosaccharide composition. (**a**) HPIEC of monosaccharide standards; (**b**) HPIEC of AP; (**c**) HPIEC of SAP. Peaks: (1) fucose; (2) galactosamine; (3) rhamnose; (4) arabinose; (5) glucosamine; (6) galactose; (7) glucose; (8) glucosamine; (9) xylose; (10) mannose; (11) fructose; (12) ribose; (13) galacturonic acid; (14) guluronic acid; (15) glucuronic acid; (16) mannose acid.

**Figure 8 foods-11-00737-f008:**
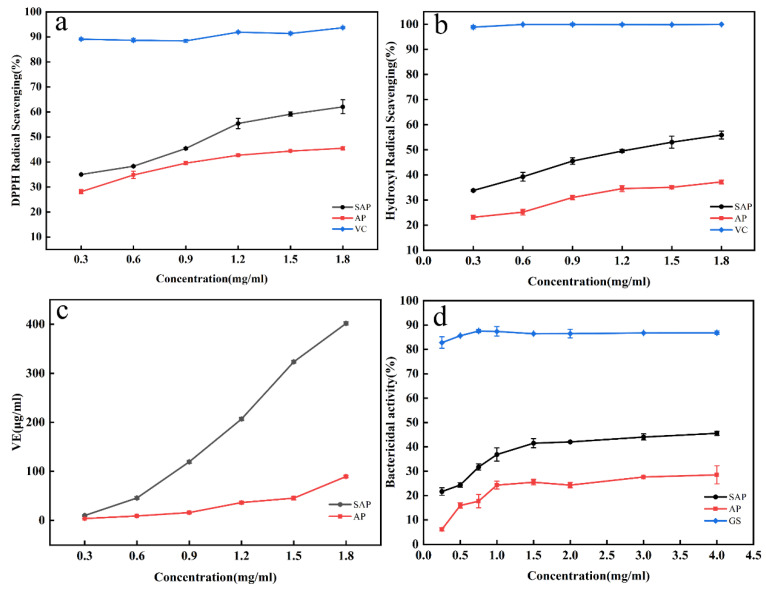
Biological activity. (**a**) Scavenging activity of AP, SAP, and VC at different concentrations for DPPH radicals. (**b**) Scavenging activity of AP, SAP, and VC at different concentrations for hydroxyl radicals. (**c**) The reducing power activity of AP and SAP at different concentrations. (**d**) Bactericidal activity of AP, SAP, and GS at different concentrations.

**Figure 9 foods-11-00737-f009:**
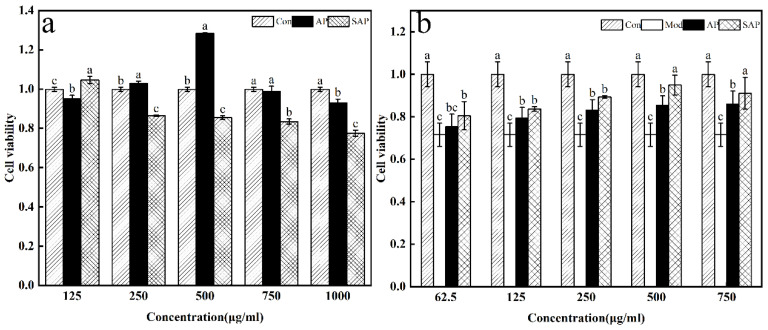
Cell viability. (**a**) Effect of different concentrations of AP and SAP on cell viability. (**b**) Effect of different concentrations of AP and SAP on cell viability of oxidatively stressed cells. Atthe same concentration, different letters in the graph indicate significant differences (*p* < 0.05), and the same letters indicate no significant differences (*p* > 0.05).

**Figure 10 foods-11-00737-f010:**
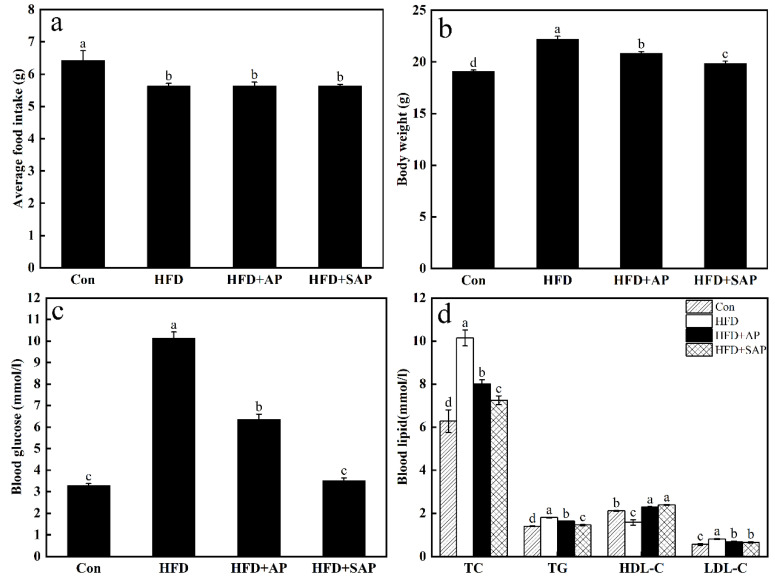
Animal experiments. (**a**) Effect of AP and SAP on average food intake. (**b**) Effect of AP and SAP on body weight. (**c**) Effect of AP and SAP on blood glucose. (**d**) Effect of AP and SAP on blood lipid.

**Table 1 foods-11-00737-t001:** Independent variables and their levels used in the response surface design.

Independent Variable	Symbol		Factor Level		
	Coded	Uncoded	−1	0	1
Reaction temperature (°C)	y_1_	X_1_	40	50	60
Ratio of CSA to PYR (mL/mL)	y_2_	X_2_	0.5	1.25	2
Reaction time (h)	y_3_	X_3_	2	2.5	3

**Table 2 foods-11-00737-t002:** Response surface central composite design, and experimental and predicted responses.

Experiment	A:temperature (°C)	B: The Ratio of CSA to PYR (mL/mL)	C: Time (h)	Reality DS	Predicted DS
1	50	1.25	2.5	0.683	0.688
2	50	1.25	2.5	0.701	0.688
3	50	0.5	3	0.542	0.552
4	60	1.25	2	0.682	0.666
5	40	0.5	2.5	0.573	0.560
6	50	1.25	2.5	0.691	0.688
7	60	1.25	3	0.486	0.477
8	40	1.25	2	0.471	0.470
9	40	1.25	3	0.513	0.519
10	50	1.25	2.5	0.708	0.688
11	60	0.5	2.5	0.635	0.638
12	50	0.5	2	0.583	0.579
13	60	2	2.5	0.632	0.643
14	50	1.25	2.5	0.681	0.688
15	50	2	3	0.489	0.492
16	40	2	2.5	0.572	0.568
17	50	2	2	0.666	0.653

**Table 3 foods-11-00737-t003:** Analysis of variance results of model.

Source	Sum of Squares	df	Mean Square	*F*-Value	*p*-Value
Model	0.1093	9	0.0121	50.77	<0.0001
A	0.0117	1	0.0117	48.95	0.0002
B	0.0001	1	0.0001	0.3534	0.5709
C	0.0173	1	0.0173	72.34	<0.0001
AB	1.0 × 10^−6^	1	1.0 × 10^−6^	0.0042	0.9502
AC	0.0142	1	0.0142	59.22	0.0001
BC	0.0046	1	0.0046	19.34	0.0032
A²	0.0156	1	0.0156	65.31	<0.0001
B²	0.0035	1	0.0035	14.71	0.0064
C²	0.0371	1	0.0371	155.26	<0.0001
Lack of Fit	0.0011	3	0.0003	2.82	0.1708

**Table 4 foods-11-00737-t004:** Physicochemical properties of the polysaccharide samples.

Physicochemical Properties	AP	SAP
Color observation	white	light yellow
Solubility	soluble	easily soluble
Carbohydrate (%)	91.45%	71.54%
Protein (%)	0.11%	0.024%
S (%)	none	9.82%
DS	none	0.724

## Data Availability

The datasets generated for this study are available on request to the corresponding author.

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
