# Peer review of "Preparation of Highly Substituted Sulfated Alfalfa Polysaccharides and Evaluation of Their Biological Activity"

_foods, 2022, doi:10.3390/foods11050737_

Round 1
Reviewer 1 Report
I have studied the work thoroughly and I come to the following remarks:
- The paper is generally good.
- The hypothesis of this work is not completely clear to me. The objectives of the study should be defined.
- you have attempted to improve alfalfa polysaccharides via the chlorosulfonic acid (CSA)-pyridine (Pyr) method to increase their biological activity. Where is the practical recommendation or is further research needed?
- What statistical model did you use when using the one-way Anova method? How you confirmed that results were normal?
- On line 74, the authors did not mention what equipment is used to dry the alfalfa at 105 °C.
- In line 75 it is better to indicate the diameters of the sieve.
- In line 165, is the abbreviation HPIC correct or do you mean HPLC?
- There are a few notes in the reference list:
- References 3, 7, 8, 9, 13, 17, 18, 24, 26, 27, 29, 33, 34, 35, 36, 41, 42, 45, 46, 50, 51 and 52 do not have a page number at the end of the studies.
- In the reference number 9 the date is repeated, why?
- Reference 11 must be spelled correctly as the author's family name was not written first.
I hope that these advices will be taken into account
I wish you much success
Author Response
Dear Reviewer,
Thank you for your response and further comments. There is no doubt that these comments are valuable and very helpful for revising and improving our manuscript. We revised the manuscript according to your comments. We answer the issues raised in turn:
- The hypothesis of this work is not completely clear to me. The objectives of the study should be defined.
Answer: Thank you for your advice. Alfalfa is a common plant with abundant production and contains a variety of active substances. Alfalfa polysaccharide is one of the important active substances. Alfalfa is suitable as a raw material for plant extracts. However, the activity of alfalfa polysaccharides was not high. This resulted in a low utilization of alfalfa polysaccharides. The objective of the study was to improve the biological activity of alfalfa polysaccharides by chemical methods. Sulfation modification has long been considered a simple and effective method to enhance biological activity. The test conditions had a significant effect on the modification results. The literature shows that the higher the content of sulfate groups of modified polysaccharides, the stronger the polysaccharide activity [1-3]. Therefore, we hope to optimize the experimental conditions by response surface testing to obtain alfalfa polysaccharides with high degree of substitution. Meanwhile, it is not clear about the changes in the biological activity of alfalfa polysaccharides after modification. We need to evaluate its biological activity. This will help us to use the modified alfalfa polysaccharide in a rational way. We also hope to provide a reference for others to study alfalfa polysaccharides or similar plant extracts.
- Li, H.; Xin, X.; Xiong, Q.; Yu, Y.; Peng, L. C. Sulfated modification, characterization, and potential bioactivities of polysaccharide from the fruiting bodies of Russula virescens. Int J of Biol Macromol, 2020, 154, 1438-1447.
- Zhao, B.T.; Tao, F.Q.; Wang, J.L.; Zhang, J. The sulfated modification and antioxidative activity of polysaccharides from Potentilla anserine L, New Journal of Chemistry. 2020, 44, 4726-4735.
- Gunasekaran, S.; Govindan, S.; Ramani, P.Sulfated modification, characterization and bioactivities of an acidic polysaccharide fraction from an edible mushroom Pleurotus eous (Berk.) Sacc, Heliyon. 2021, e05964, 1-11
- You have attempted to improve alfalfa polysaccharides via the chlorosulfonic acid (CSA)-pyridine (Pyr) method to increase their biological activity. Where is the practical recommendation or is further research needed?
Answer: Thank you for your comments to our manuscript. Our results show that sulfated alfalfa polysaccharide has good antioxidant and antiobesity ability. Suitable concentrations of alfalfa polysaccharide sulfate have the potential to improve intestinal permeability. We inferred that sulfated alfalfa polysaccharide has the function of treating chronic inflammation due to obesity. We will conduct further in vivo studies in animals to determine its actual effects and mechanism of action.
- What statistical model did you use when using the one-way Anova method? How you confirmed that results were normal?
Answer: Thank you for your comments. Our experiments were performed in replicates of three or more. Data was collected and collated using Microsoft Excel table. Data were analyzed by using SPSS software. DPPH radicals, hydroxyl radicals, reducing power activity and antimicrobial capacity were performed as one-way ANOVAs for different concentrations within the same group. Cell viability assay and animal experiments were performed as one-way ANOVAs for different group within the same concentration. All data were analyzed by one way ANOVA followed by LSD.
- On line 74, the authors did not mention what equipment is used to dry the alfalfa at 105 °C.
Answer: We thank the reviewer for pointing out this oversight. We used ovens to dry alfalfa. We have added this message in manuscript.
- In line 75 it is better to indicate the diameters of the sieve.
Answer: Thank you for your advice. We used 100-mesh sieves. We have added the specifications of the sieve in the manuscript.
- In line 165, is the abbreviation HPIC correct or do you mean HPLC?
Answer: Thank you for your comments. HPIC is High performance ion chromatography
- There are a few notes in the reference list: References 3, 7, 8, 9, 13, 17, 18, 24, 26, 27, 29, 33, 34, 35, 36, 41, 42, 45, 46, 50, 51 and 52 do not have a page number at the end of the studies.
Answer: We thank the reviewer for pointing out this oversight, and have now added page number at the end of the all reference.
- In the reference number 9 the date is repeated, why?
Answer: We apologize to the reviewer for the error, the reference number 9 was edited incorrectly. We’ve corrected the reference in manuscript.
- Reference 11 must be spelled correctly as the author's family name was not written first.
Answer: We thank the reviewer for pointing out this oversight, and have now corrected this point in manuscript.
Thank you again for your positive and constructive comments and suggestions on our manuscript. We hope you will find our revised manuscript acceptable for publication.

Reviewer 2 Report
The manuscript submitted to Foods for publication by Li et al., titled: "Preparation of highly substituted sulfated alfalfa polysaccharides and evaluation of their biological activity" is aiming to evaluate the activity of highly substituted sulfated polysaccharides.
The manuscript addresses an interesting topic, the language and writing style are good although it would be suggested for an English native speaker to have the manuscript proofread.
The reviewer would like to note the following points towards improving the manuscript.
- The authors are making the point on the biological activity for the AP. It is important to point out that biological activity in a system is not necessarily linear as it may be involved in signaling for example in which case not significant amounts are required. Furthermore, there are numerous examples of compounds which are harmful beyond a certain concentration/dose while beneficial prior to reaching that level. This would need to be discussed in the discussion section.
- Have the authors considered potential side-effects of sulfation in vivo?
Author Response
Dear Reviewer,
Thank you for your response and further comments. There is no doubt that these comments are valuable and very helpful for revising and improving our manuscript. We revised the manuscript according to your comments. We answer the issues raised in turn:
- The authors are making the point on the biological activity for the AP. It is important to point out that biological activity in a system is not necessarily linear as it may be involved in signaling for example in which case not significant amounts are required. Furthermore, there are numerous examples of compounds which are harmful beyond a certain concentration/dose while beneficial prior to reaching that level. This would need to be discussed in the discussion section.
Answer: Thank you for your advice. In our cell activity assay, we also found this phenomenon. Above a certain concentration of drug, the cellular activity decreased, while before this concentration, the cellular activity increased with increasing concentration. This phenomenon suggests that high drug concentrations can damage cells. There is an upper tolerance limit of organisms to sulfated alfalfa polysaccharide and alfalfa polysaccharide. We have enhanced the discussion of this phenomenon in the manuscript.
- Have the authors considered potential side-effects of sulfation in vivo?
Answer: Thank you for your comments. We have observed a significant improvement in the biological activity of sulfated alfalfa polysaccharide compared to alfalfa polysaccharide. Related literature suggests that polysaccharides cause changes in the gut microbial community [1,2]. In vitro tests have found that alfalfa polysaccharide sulfate has a stronger effect on bacteria than alfalfa polysaccharide. This implies that alfalfa polysaccharide sulfate may not exert the same effect in the intestine as alfalfa polysaccharide. However, we cannot be sure whether this effect is beneficial to the host and further in vivo tests are needed to investigate it. Meanwhile, whether the polysaccharide with the sulfate group attached could digested and broken down in the host and the changes in the in vivo microenvironment caused also deserve further consideration. We will continue to study this.
[1] Chang, C, J.; Lin, C, S.; Lu, C, C. et al. Ganoderma lucidum reduces obesity in mice by modulating the composition of the gut microbiota[J]. Nature Communications, 2017, 6:7489.
[2] Shannon, E.; Conlon, M.; Hayes, M. Seaweed Components as Potential Modulators of the Gut Microbiota[J]. Marine Drugs, 2021, 19,58-408.
Thank you again for your positive and constructive comments and suggestions on our manuscript. We hope you will find our revised manuscript acceptable for publication.

Reviewer 3 Report
04 Feb -2022
Journal: Foods
Title: Preparation of highly substituted sulfated alfalfa polysaccharides and evaluation of their biological activity
Dear Editor:
The authors have investigated the synthesis of Alfalfa polysaccharides (AP) and evaluate its activity as antioxidant, antibacterial and antiproliferation. The manuscript carries the scientific merit and would be suitable for publication after major revision as suggested below.
Nermeen Yosri, PhD
Comments to authors:
- The graphical abstract is highly recommended.
- The authors would add the originality of the work in the abstract
- Could you add a brief introduction about Alfalfa polysaccharides in the abstract?
- Could you add the structure of Alfalfa polysaccharides?
- Please add some of photos of experiments to material and methods section?
- " With slight modifications"; please give more details
- "CSA-Pyr method"; indicate for what?
- What does VC stand for?
- Please specify the negative and positive controls for each experiment
- "Staphylococcus aureus (S. aureus)" should be italic
- (800 mg/kg)"; why the authors choose that conc; they should used at least 3 different concentrations
- Could you explain in more details why is SAP is more active than AP in all activities?
- Please add IC50 and MIC of the biological activities
- The discussion part could be more focused.
- The conclusion section should be reduced with focusing on the main outcomes of the study.
- The authors could benefit from the following reference:
Yosri N, Khalifa SA, Guo Z, Xu B, Zou X, El-Seedi HR. Marine organisms: Pioneer natural sources of polysaccharides/proteins for green synthesis of nanoparticles and their potential applications. International Journal of Biological Macromolecules. 2021 Dec 15;193: 1767-98.
Taken together:
- The authors would unify the style of the references based on the journal instructions.
- English editing is highly required.
- Please, re-check the punctuations, syntax, and English grammar throughout the manuscript.
Author Response
Dear Reviewer,
Thank you for your response and further comments. There is no doubt that these comments are valuable and very helpful for revising and improving our manuscript. We revised the manuscript according to your comments. We answer the issues raised in turn:
- The graphical abstract is highly recommended. The authors would add the originality of the work in the abstract
Answer: Thank you for your suggestions. We think this is a very good suggestion. We presented the technical lines and conclusions of the experiment in a graphical summary and added it to the manuscript.
- Could you add a brief introduction about Alfalfa polysaccharides in the abstract?
Answer: Thank you for your suggestions. Due to the word limit of the abstract, we can only give a short description of alfalfa polysaccharides in the abstract. In the introduction we present the current status of research on alfalfa polysaccharides.
- Could you add the structure of Alfalfa polysaccharides?
Answer: I am sorry that we did not investigate the structure of alfalfa polysaccharides and the substitution sites of sulfate groups. The purpose of our study was to obtain highly substituted alfalfa polysaccharides and to explore the biological activity of sulfated alfalfa polysaccharides. Both alfalfa species and growth period affect the molecular structure of alfalfa polysaccharides. The main structures of alfalfa polysaccharides are not affected by differ significantly between species or growth periods, but there were differences in molecular weight and monosaccharide composition ratios. Therefore, we only investigated the effect of sulfation modification on the molecular weight and monosaccharide composition of alfalfa polysaccharides.The following literature can be consulted for the structure of alfalfa polysaccharides [1-2].
[1] Zhang, C.; Li, Z. M.; Zhang, C. Y.; Li, M. M.; Lee, Y.; Zhang, G. G. Extract methods, molecular characteristics, and bioactivities of polysaccharide from alfalfa (Medicago sativa L.), Nutrients. 2019, 11, 1181-1196.
[2] Liu, X, G.; Xu, S, S.; Ding, X, D.; Yue, D, D.; Bian, J.; Zhang, X.; Zhang, G, L.; Gao, P, Y.; Structural characteristics of Medicago Sativa L. Polysaccharides and Se-modified polysaccharides as well as their antioxidant and neuroprotective activities, Int J of Biol Macromol, 2020, 147, 1099-1106.
- Please add some of photos of experiments to material and methods section?
Answer: According to your suggestion, we have added photos of alfalfa polysaccharide and sulfated alfalfa polysaccharide.
- " With slight modifications"; please give more details
Answer: We have refined the test method.
- "CSA-Pyr method"; indicate for what?
Answer: CSA-Pyr method indicate for chlorosulfonic acid (CSA)-pyridine (Pyr) method. The content can be found in line 16.
- What does VC stand for? Please specify the negative and positive controls for each experiment
Answer: Thank you for your comments. VC is a common antioxidant and was used as a positive control group in the antioxidant experiment. The rest of the trials have also indicated their respective control groups.
- "Staphylococcus aureus (S. aureus)" should be italic
Answer: We thank the reviewer for pointing out this oversight, we have corrected this error.
- (800 mg/kg)"; why the authors choose that conc; they should used at least 3 different concentrations
Answer: Thank you for your comments. The concentration was determined based on relevant literature [3]. The concentration was determined to be high for small animal after testing in the literature. The purpose of this test section is to determine if mice can tolerate the same high concentration of alfalfa polysaccharide sulfate. Next, we will design in vivo tests with three concentrations, low, medium and high, using this concentration as an upper limit and investigate the effects and mechanisms of different concentrations of sulfated alfalfa polysaccharide in vivo.
[3] Zhang C.Y.; Gan L.P., Du M.Y.; Shang Q.H.; Xie Y.H.; Zhang G.G. Effects of dietary supplementation of alfalfa poly-saccharides on growth performance, small intestinal enzyme activities, morphology, and large intestinal selected microbiota of piglets, Livest Sci. 2019, 223, 47-52.
- Could you explain in more details why is SAP is more active than AP in all activities?
Answer: Thank you for your comments. This phenomenon can be explained in three aspects.
- The sulfation modification allows the polysaccharide to carry a large number of electrons. Electrons can abort chain reactions and scavenge free radicals. Therefore, the antioxidant activity of alfalfa polysaccharide sulfate is stronger than that of alfalfa polysaccharide.
- The sulfate group is a hydrophilic group. Alfalfa polysaccharide sulfate carries a large number of sulfate groups. The water solubility of the polysaccharide is enhanced, which facilitates the effect of the polysaccharide.
- Several studies have shown that polysaccharides can alleviate obesity by modifying the structure of the intestinal microbiota [4-5]. Alfalfa sulfate polysaccharide should have a similar ability. Alfalfa polysaccharide sulfate may also reduce intestinal permeability by promoting the proliferation of intestinal epithelial cells, which may help alleviate chronic inflammation caused by obesity. Therefore, sulfated alfalfa polysaccharide has the ability to reduce obesity.
The above has been added to the discussion of each experiment.
[4] Chang, C, J.; Lin, C, S.; Lu, C, C. et al. Ganoderma lucidum reduces obesity in mice by modulating the composition of the gut microbiota[J]. Nature Communications, 2017, 6:7489.
[5] Shannon, E.; Conlon, M.; Hayes, M. Seaweed Components as Potential Modulators of the Gut Microbiota[J]. Marine Drugs, 2021, 19,58-408.
- Please add IC50 and MIC of the biological activities
Answer: Sorry for this. Sulfation modification improved the bacterial inhibitory ability of alfalfa polysaccharides. But the inhibitory ability of alfalfa polysaccharide and alfalfa polysaccharide sulfate was weak, and the values of IC50 and MIC could not be obtained. According to the literature, we found that some polysaccharides have the ability to inhibit bacteria. Therefore, we determined the antibacterial ability of alfalfa polysaccharide and alfalfa polysaccharide sulfate in our experiment. However, the test results showed that alfalfa polysaccharides had poor antibacterial ability. Compared with alfalfa polysaccharide, the antibacterial ability of alfalfa polysaccharide sulfate was improved, but not as expected. The IC50 value calculated by formula was not reasonable. The bacterial inhibition rate of SAP was less than 50%. We also found that bacteria developed resistance to alfalfa polysaccharide and alfalfa polysaccharide sulfate with increasing concentrations. Therefore, sulfation modification could improve the bacterial inhibitory ability of alfalfa polysaccharide. But the bacterial inhibitory ability of alfalfa polysaccharide sulfate could not support its use as a potential antibacterial drug. The weak antibacterial ability of alfalfa polysaccharide limits the upper limit of sulfated alfalfa polysaccharide.
- The discussion part could be more focused.
Answer: Thank you for your suggestions. We have optimized the content of the discussion section.
- The conclusion section should be reduced with focusing on the main outcomes of the study.
Answer: Thank you for your suggestions. We simplified the content of the conclusion section. The focus is placed on the discussion section.
- The authors would unify the style of the references based on the journal instructions.
English editing is highly required.
Answer: According to your requirements. We have edited and corrected the references, added missing page numbers and corrected punctuation.
- Please, re-check the punctuations, syntax, and English grammar throughout the manuscript.
Answer: According to your requirements. We have completed checking all aspects of the revised article.
Thank you again for your positive and constructive comments and suggestions on our manuscript. We hope you will find our revised manuscript acceptable for publication.

Round 2
Reviewer 2 Report
The authors have addressed the reviewers' comments reasonably.
Reviewer 3 Report
Dear Editor
Yes, it has been modified according to our suggestions.
I recommend the paper for publication.
Kindest regards, Nermeen